# Description of e-Health Initiatives to Reduce Chronic Non-Communicable Disease Burden on Brazilian Health System

**DOI:** 10.3390/ijerph181910218

**Published:** 2021-09-28

**Authors:** Daniela Laranja Gomes Rodrigues, Gisele Silvestre Belber, Igor da Costa Borysow, Marcos Aurelio Maeyama, Ana Paula Neves Marques de Pinho

**Affiliations:** 1Social Responsibility Department, Hospital Alemão Oswaldo Cruz, São Paulo 01323-020, Brazil; gsbelber@haoc.com.br (G.S.B.); iborysow@haoc.com.br (I.d.C.B.); ampinho@haoc.com.br (A.P.N.M.d.P.); 2Telehealth Center, Florianópolis 88040-900, Brazil; marcos.aurelio@univali.br

**Keywords:** chronic non-communicable diseases, unified health system, primary health care, Brazil

## Abstract

Chronic non-communicable diseases (NCD) account for 72% of the causes of death in Brazil. In 2013, 54 million Brazilians reported having at least one NCD. The implementation of e-Health in the Unified Health System (SUS) could fill gaps in access to health in primary health care (PHC). Objective: to demonstrate telehealth strategies carried out within the scope of the Institutional Development Support Program of the Unified Health System (PROADI-SUS) and developed by Hospital Alemão Oswaldo Cruz, between 2018 and 2021, on evaluation, supply, and problem-solving capacity for patients with NCDs. Methodology: a prospective and descriptive study of three projects in the telehealth areas, using document analysis. The Brasil Redes project used availability, implementation, and cost-effectiveness analysis, TELEconsulta Diabetes is a randomized clinical trial, and Regula Mais Brasil is focused on the waiting list for regulation of specialties. All those strategies were developed within the scope of the SUS. Results: 161 patients were attended by endocrinology teleconsultation in one project and another two research projects, one evaluating Brazil’s Telehealth Network Program, and another evaluating effectiveness and safety of teleconsultation in patients with diabetes mellitus referred from primary care to specialized care in SUS. Despite the discrepancy in the provision of telehealth services in the country, there was an increase in access to specialized care on the three projects and especially on the Regula Mais Brasil Collaborative project; we observed a reduction on waiting time and favored distance education processes. Conclusion: the three projects offered subsidies for decision-making by the Ministry of Health in e-Health and two developed technologies that could be incorporated into SUS.

## 1. Introduction

With the change in the epidemiological and demographic profile of the population, in addition to the growing increase in chronic pathologies and people in vulnerable situations, the need for an integrated approach to the individual concerning health care emerges [1,2,3]. Chronic non-communicable diseases (CNCD) are described as diseases that involve the circulatory system, diabetes, cancer, and chronic respiratory diseases. This group of diseases usually occurs more commonly in developing countries, accounting for 63% of the causes of death in the world [4,5] and corresponding to one-third of deaths in people under 60 years of age [6]. The last national health survey conducted by the Brazilian government in 2013 [7] showed that 72% of the causes of death were related to CNCD, that is, by diseases that could have their impact reduced if health strategies focused mainly on primary care were implemented.

Digital health, or e-Health, can be defined as “the safe and cost-effective use of information and communication technologies in support of health and health-related fields”, with the World Health Organization considering making e-Health a global priority for health systems development [8]. The implementation of e-Health can substantially contribute to filling some gaps in the performance of primary health care (PHC), especially in developing countries such as Brazil [9]. One of these strategies is to expand the use of telemedicine to provide diagnostic and consulting support to professionals in distant locations, supporting the physician in such locations and improving the skills and knowledge of remote healthcare providers [10].

The distribution of income and access of the Brazilian population to technological resources is characterized by geographical and social inequality. Even so, the report “Digital 2020”, published by *We Are Social* in January 2020, identified 150.4 million internet users in Brazil [11], accounting for an increase of 6.0% (8.5 million) when compared to the same month of the previous year, and by June 2020 there were identified 234 million active smartphones, a number that has been larger than the Brazilian population since 2017, according to the 31st edition of the Getúlio Vargas Foundation (FGV) Annual Research on the IT Usage [12]. Considering the public health area, there is a partnership between the Ministry of Health and the Ministry of Science, Technology, and Communications to provide internet access in 100% of health care units in Brazil [13]. That way, we can recognize that there is an increasingly favorable context for the wide deployment of telemedicine in the country. Despite this, there are still challenges and important aspects to be considered, for example, a Brazilian resolution that allows greater use of telehealth strategies throughout the country [14,15].

This article aims to describe telehealth strategies carried out in the years 2018 to 2021 by the Hospital Alemão Oswaldo Cruz, in partnership with the Brazilian Ministry of Health through the Support Program for Institutional Development of the Unified Health System (PROADI-SUS) [16,17].

## 2. Materials and Methods

Between 2018 and 2021, Hospital Alemão Oswaldo Cruz, in partnership with the Ministry of Health, through PROADI-SUS, developed projects and actions in Telehealth aimed at improving management in various spheres of the SUS. Of these, we review three strategies aimed at reducing the impact of CNCD on health using different strategies in e-Health for this purpose. As these are three different projects, we will describe the methodology of each one separately.

### 2.1. Brasil Redes Project

The Brasil Redes Project aimed to carry out a diagnostic evaluation of the National Telehealth Brazil Networks Program (PNTBR–in Portuguese), which completed 10 years of existence in 2017 [18]. This program aims to promote strategies to support primary health care through distance communication actions that are carried out by telehealth centers. As of 2011, the main activities to be carried out by the centers were as defined: teleconsulting, telediagnosis, tele-education, and training second opinion [19].

Five studies were carried out for this evaluation: evaluability study, implementation evaluation, and cost-effectiveness evaluation of two services offered in the list of activities foreseen by the program and program cost analysis [20]. The evaluability study was the first stage, aiming to identify the resources, the stakeholders, the processes carried out, the evaluative questions, and the expectations regarding the results of the referred program. This study included data collection work between the months of August and December 2018. The second study was the implementation evaluation, which had two major stages: socio-historical analysis of the genesis of the program and the implementation evaluation itself. The economic evaluations addressed the teledermatology service of the Telehealth Center of Santa Catarina [21] and the telecardiology service of the Telehealth Center of the Hospital das Clínicas of the Federal University of Minas Gerais [22] (Figure 1). The last study evaluated the unit value of the electrocardiogram and Holter report to be implemented as a procedure made available to the population by the Unified Health System.

Table 1 outlines a systematization of the five studies carried out.

### 2.2. Regula Mais Brasil Colaborativo

Regula Mais Brasil Collaborative [17] is a project developed to qualify the outpatient care regulation process (evaluation of referrals to specialized care) using telehealth technologies, with an evaluation of cases in regulation, use of access protocols to specialties, and teleconsulting support for case resolution. Furthermore, this project monitors the waiting period process, where the patient is registered in the online system where he awaits the emergence of a vacancy for the requested specialty. Usually, this waiting time is 1 to 2 years.

Motivated by the COVID-19 pandemic [15], the Ministry of Health of Brazil authorized teleconsultations to face the pandemic, which had been prohibited in Brazil until then, and invited the Hospitals to participate in the Regula Mais Brasil Collaborative to offer teleconsultations, thus expanding the scope of the project. The purpose of this offer was to reduce the risk of transmission of the virus, both in displacement and in the environment of health services, to avoid contact with possible contaminants, and to maintain the care of acute and chronic diseases in an off-site manner. In addition, the importance of this offer was enhanced by the closing of several elective outpatient care services at the beginning of the pandemic, which could lead to the worsening of cases due to a lack of assistance.

### 2.3. Teleconsulta Diabetes

This project’s main objective, until in recruitment phase, is to conduct clinical research to test the hypothesis that teleconsultation is non-inferior to the face-to-face care of patients with diabetes mellitus (DM) type II referred from primary health care (PHC) to specialized care in SUS [23].

This is a randomized, pragmatic, phase-2, single-center, open, non-inferiority clinical trial, with central randomization, allocation confidentiality, and data analyst blinding, to assess the efficacy and safety of specialized care by teleconsultation compared to face-to-face care. A total of 250 participants of both sexes over 18 years old, with type 2 diabetes mellitus, will be included. This sample size will allow evaluating the non-inferiority of up to 0.5% between groups, assuming a standard deviation of 1.30.

The outcomes that will be analyzed in this study are: the mean difference in the percentage of glycated hemoglobin (HbA1c) post- and pre-intervention in patients diagnosed with type 2 DM in 3 and 6 months, fasting blood glucose measurements and blood count, measurements of serum urea and creatinine, lipid profile measurements, systolic pressure measurements, in-office measurements, in-office diastolic pressure measurements, measurements of body weight and body mass index (BMI), the incidence of any adverse events, and quality measurements of patients’ lives using the diabetes quality of life measurement questionnaire (DQOL) [24].

In the micro-costing analysis [25], this study performs the definition of the consumption time equations of each resource per product/service and extrapolation of the findings to define the productive capacities of the basic health units (UBS–in Portuguese) through real care data by the flow of regulation in the city of Joinville.

## 3. Results

The main target audience of the Telessaúde Brasil Redes National Program is primary health care teams, and with that many of the actions offered by the centers were focused on caring for people with CNCD. In this sense, the project results highlighted opportunities for improvement of the telehealth program for CNCD at the federal level. In Brazil’s public health system, there is a necessity to homogenize the financing values considering the production potential of the centers, mainly for telediagnosis, and monitoring and evaluation methods of results. For the spheres of state centers aiming to value their actions, there is a need to reflect on challenges in the use of these services by health workers and on regional partnerships. The set of methods offered a broad vision of the program and articulated more detailed analyses of certain services offered by the telehealth centers. The purpose was to offer subsidies to the Ministry of Health in improving the telehealth program and using the expertise of certain centers as inspiring the formulation of procedures used to be offered by SUS.

Since the creation of the Brazil Redes program in 2007, 33 telehealth centers have been identified in various regions of the country (Figure 1), with very diversified health access strategies, as described in Table 2.

In this sense, the centers sought to articulate different fronts of activities to contemplate permanent education and the improvement of primary care professionals, whether through courses, virtual lectures, teleconsulting, production of training second opinions, and telediagnosis. Regarding the processes evaluated in the economic component, the two studies carried out with the Telehealth Center at Clinic’s Hospital from the Federal University of Minas Gerais clarify the cost and budget impact forecast for the implementation of telecardiology services in the public service network, which can greatly contribute to the improvement of care for people with CNCD.

The Hospital Alemão Oswaldo Cruz carried out consolations in the Regula Mais Brasil Collaborative Project via teleconsultation of users, in the specialties of neurology, endocrinology, orthopedics, and mental health, totaling 1097 teleconsultations in this period. Specific actions for CNCDs are concentrated in the specialty of endocrinology, in which 161 teleconsultations were carried out, all with SUS users in the city of Recife, city located in the state of Pernambuco, in northeastern Brazil (Figure 1). The results obtained in this project are displayed in Table 3.

Depending on the prevalence of endocrinological diseases, diabetes mellitus appears as the most prevalent reason for teleconsultations performed. Regarding the technology used for contact between the doctor and the patient, video calling (audio and video) was mostly used, since it represents the best possibility of remote interaction. The exclusive audio resource was only used when it was not possible to use video calling.

The outcomes of the last teleconsultation were performed to allow an assessment of the resolvability potential of this assistive technology. The follow-up outcomes in PHC (40.3%) represent cases in which the specialist, based on the assessment in teleconsultation, considers that, based on the complexity of the case, the patient can be followed up in the PHC, close to their residence, avoiding unnecessary referral and displacement of the patient to the specialized center. In addition, he receives the first guidance and conduct by the specialist via teleconsultation. Likewise, the reassessment in teleconsultations (23.4%) also represents the cases in which digital technology allows for adequate follow-up with a specialist, without the need to travel. On the other hand, cases in which the outcomes indicate the need for a face-to-face consultation with the specialist, whether immediate or elective, represent the limitation of teleconsultation in the assessment or follow-up of cases. It is worth mentioning that all cases evaluated did not have a first face-to-face consultation which, in some cases, limits a better diagnosis.

What corroborates the potential of teleconsultations is the result of the Net Promoter Score [26], which assesses user satisfaction concerning the service, and which, on a scale from 0 to 100, presented an index of 93 points in teleconsultations in endocrinology carried out and evaluated by the project patients.

Regarding the results of the teleconsultation diabetes research project, it was not yet possible to demonstrate aspects of the research results related to the results of complementary exams, questionnaires measuring the quality of life in diabetes (DQOL), and satisfaction assessment with telemedicine, due to the small number of participants included so far. Of the patients included in the study, the mean age was 59.4 years, with a predominance of 78% of white people, followed by black (12%) and mixed race (10%), and a slight predominance of males (54%) of the included participants.

In addition, within the scope of the teleconsultation project, the time-driven activity-based costing (TDABC) micro-costing methodology, or costs based on time and activity, was adopted because it allows the identification of the unit cost of the service within the expected efficiency conditions, also acting as a comparison metric. In this scenario, the first hypothesis considered the duration of the teleconsultation and the face-to-face consultation to be identical, as well as the duration of the pre-consultation activities. This is a preliminary conclusion, considering that recruitment, until this period, was minimal and with low statistical power to detect differences in times. The second hypothesis considered that the costs with materials and structure of the UBS’s and the Polyclinic are also identical since the Health Department of Joinville is in the middle of the process of changing the methodology for this type of cost and must provide these data in a manner more accurate by mid-2021. In this hypothetical scenario, the cost of teleconsultation was 5% higher than that of a face-to-face consultation. As in both cases, the consultation is responsible for more than 90% of the total cost, the difference in duration between the two types of consultation will be the determining factor for the difference in costs between them. More details that relate the methodologic and statistic description are outlined in the design paper published before [23].

## 4. Discussion

Teleconsultation is a medical act and must abide by the Brazilian Medical Code of Ethics (MCE) [27,28,29]. As provided in Article 37 of the MCE, the physician is prohibited from “prescribing treatment or other procedures without direct examination of the patient, except in cases of urgency or emergency and proven impossibility of performing it, and, in such circumstances, must do so immediately after ceasing the impediment” [30]. The State of Emergency in Public Health of National Importance (ESPIN) [31] triggered during the SARS-CoV-2 pandemic fits into this prerogative, thus supporting teleconsultations during this exceptional period where the Brazilian society had the opportunity to experience remote medical care more extensively.

Critics of telemedicine approval warn that the possible overuse and careless use of telemedicine could turn doctors into “telemarketing operators”, which could lead to poor quality in appointments, medical errors, and unemployment by reducing the number of face-to-face doctors [32]. Although the movements of criticism to teleconsultation caused a fanfare in the mainstream media, they do not seem to reproduce what the Brazilian medical professional wants. A survey carried out in February 2020 by the São Paulo Physicians Association (APM) with over 2200 physicians from 55 different specialties revealed that 64.39% of doctors wanted a regulation that would allow the expansion of services and assistance to the population, including direct doctor-to-patient teleconsultation [33].

This whole context evidences the lack of alignment between the federal government, the Brazilian Medical Federal Council, health plan operators, medical associations, and medical professionals, especially concerning teleconsultations. The challenge is how to expand access to medical services mainly to specialists for populations in remote regions, reduce healthcare costs and the displacement of patients, and on the other hand, minimize the fear of damage to the medical profession [27]. Other challenges now faced by specialists are data insecurity, trivialization of teleconsultations, and the production of misdiagnoses and prescriptions, in addition to avoiding possible unemployment of doctors. The country should take advantage of the current situation and promote a wider discussion on the benefits and limitations of permanent and full permission to use telemedicine, bringing to the agenda socioeconomic, cultural, and technological issues.

The Brazil Redes program showed the presence of decentralized and disseminated strategies throughout the national territory, with actions in telehealth, focused not only on CNCDs but also other health promotion strategies. The actions of teleophthalmology with identification of diabetic retinopathy based in Mato Grosso [34,35] confirm the efficiency and potential of supporting the health of the population using the Unified Health System if the use of similar strategies were more widespread in the country. They also serve as a model as strategies to be implemented in other developing countries.

The Regula Mais Brasil project, through the implementation of teleconsultation in the Unified Health System, demonstrated, in an unprecedented way, how strategies that allow the population’s access to different medical specialties can be feasible, in addition to allowing the capillarization of care, through the strategy to take the specialist wherever the patient is and not the other way around. Especially in the period of the SARS-CoV-2 pandemic, this project can maintain access to health care, especially for users with diabetes and sequelae after a stroke, ensuring the maintenance of care [36].

Regarding the teleconsultation diabetes research project, if the efficacy and safety for patients assisted by teleconsultation are confirmed, once the normative issues are resolved by the Federal Council of Medicine (CFM), this will be able to effectively contribute to the promotion access of patients to the public health system, including specialist physicians. Likewise, there may be an increase in the resolvability of the population’s health needs, breaking the geographical barrier that a country with continental dimensions, such as Brazil, imposes on the provision and standardization of health, in addition to the potential savings for health systems, which may be used safely and with quality.

This paper has several limitations: one of then is the fact we make a compilation of three different strategies implemented at SUS in a tentative way to delivery medical assistance to the general population. As the initial projects were not designed to analyze the impact on outcomes, for example, the incidence of myocardial infarct or stroke, the authors only could initiate a discussion, showing that e-Health strategies can be performed in a public health system.

## 5. Conclusions

Although there is evidence in favor of the use of telehealth strategies to deal with CNCDs, in Brazil, due to difficulties in accessing technology in addition to important care gaps and legal impediments, we observe a lack of strategies in this area for health promotion within the scope of the SUS.

Therefore, given the above, there is a need to generate reliable scientific evidence of the effectiveness and safety of applicability within the context of regulation and access to the SUS. In this article, the authors demonstrate how specific initiatives in telehealth, through partnerships between the Ministry of Health and Hospitals of Excellence, can foster the development of new research and assistance strategies within the scope of e-Health in Brazil.

## Figures and Tables

**Figure 1 ijerph-18-10218-f001:**
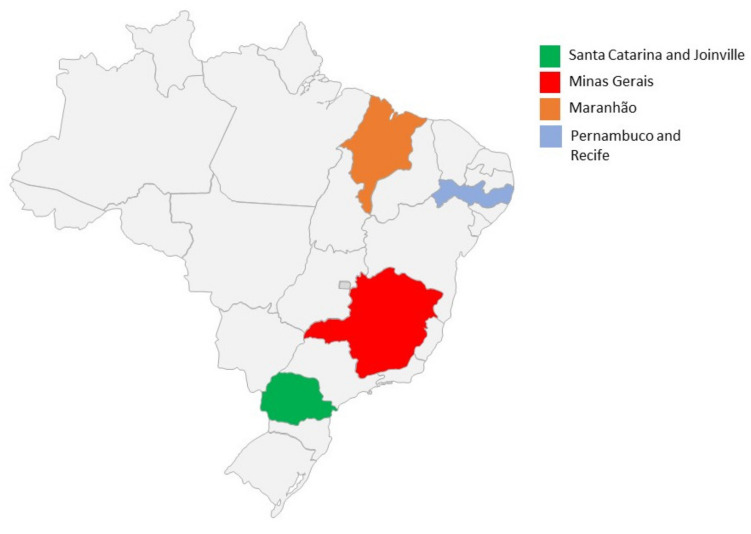
Brazil’s states with telehealth initiatives included in the current article.

**Table 1 ijerph-18-10218-t001:** Systematization of Brazil’s Rede project.

Analysis	Objective	Methodology	Institutions
Evaluability	Check the feasibility of the process Evaluative, exploring expectations and needs of the interest groups involved and the degree of organization and implementation of the initiative	Participant observation, members interviews, elaboration of a Logical Model	State telehealth centers in Maranhão, Pernambuco, Santa Catarina and 2 Telehealth Centers in Minas Gerais (Figure 1)
Implementation	Check the necessary adaptations of the proposals of the program by the centers, according to their realities (region, population characteristics, political context, history of disputes) and conformation. Identify limitations of the service network and the articulation with the main actors of the program in question.	Socio-historical study 17 semi-structured members interviews were carried out Implementation evaluation Participant observation and interviews with members of the telehealth centers and network agents Implementation degree evaluation matrix	Socio-historical study: representatives of telehealth centers across the country and former employees of the Ministry of Health
Teledermatology’s Economic Evaluation	Carry out cost-effectiveness analysis of the teledermatology service Identify the specific activities and vocations of the Nucleus, aiming at raising expenses and results obtained to feed the economic evaluation models Define the effectiveness indicators and economic evaluation models Develop economic assessment	Cost-effectiveness analysis comparing the teledermatology service with the provision of conventional care, from the society’s perspective, considering the period from January 2016 to December 2018. The analysis was developed using a tree decision, considering the possible transition to alternatives: teledermatology or conventional service.	Santa Catarina’s Telehealth Center (Figure 1)
Telecardiology’s Economic Evaluation	Cost-effectiveness analysis of the telecardiology service.	Cost-effectiveness analysis comparing the telecardiology service with the provision of conventional care, from the perspective of society, considering the period between July 2018 and June 2019 The analysis was developed using a decision tree, considering the possibilities transition to alternatives: telecardiology or conventional care	Federal University of Minas Gerais Telehealth Center
Analysis of costs and budgetary impact of the inclusion of the telecardiology service in the list of Unified Public Health System (SUS) procedures	Identify the unit payment amount, by the SUS, of an electrocardiogram and Holter report, via telehealth, with a view to its incorporation into the list of SUS procedures	Description, measurement, and analysis of the costs involved in providing the telecardiology service. Calculation of the total monthly cost and the average cost per report. Analysis of the budgetary impact arising from the incorporation of the service into the SUS.	

**Table 2 ijerph-18-10218-t002:** Contributions of the evaluative studies of the National Telehealth Brazil Networks Program to MS decision-making, with a focus on CNCD.

Analysis	Results	Contributions to the Ministry of Health in Decision Making for Chronic Non-Communicable Diseases (CNCD)
Evaluability study	Management mechanisms identified and clear definition of work proposals and expected results. Heterogeneity between health centers, including activities offered and receipt of funds. There is institutional weakness in the program’s national coordination.	In-depth analysis of the heterogeneity of the centers and their regional contexts is necessary to better plan the actions to be offered by each one, and regularize the funds transferred by the federal entity. This could improve the telehealth strategy aimed at primary care.
Implementation assessment	Through the analysis of four cores, heterogeneity in the provision of telehealth services, differences in the receipt and use of funds, and diversity in articulations with state decision-making bodies were confirmed. The federal administration did not develop adequate strategies for managing the funding and productivity of the centers. The centers raise funds from different sources and seek to keep their services running.	The actions of teleconsultation, tele-education and training second opinion are aimed at primary care and help in the training of workers in the care of CNCD. Despite the fragile measurement of results, these actions are recognized by workers as relevant to improvement. Telediagnosis in cardiology, dermatology, spirometry, stomatology, and ophthalmology demonstrate an impact on the CNCD healthcare network. Support to regulation centers organize and optimize queues for specialists and promote greater resolution in the primary care.
Teledermatology’s economic evaluation	The cost-effectiveness analysis shows that the service, using the teledermatology strategy, costs USD 191.38 per patient, while the use of the conventional service costs USD 220.68 per patient. As the performance of the service was used as an effectiveness parameter, there are no significant differences between the alternatives. If the services have equal patient care capacity, there are only differences in costs, which result in USD 59.89 more for the care of a patient in the conventional strategy compared to teledermatology.	It is concluded that conventional care is dominated by teledermatology, presenting itself as a good strategy to be implemented and/or financed by the public administration. This service will be able to contribute with greater agility in scheduling and carrying out consultations, preventing dermatological problems from evolving in severity, as well as helping people from regions farther away from the places where the specialists are located to have easier access to the dermatological examination.
Telecardiogy’s economic evaluation	The cost-effectiveness analysis shows that the service, using the telecardiology strategy, considering that all patients have their problems solved in secondary care, cost USD 47.35 purchasing power parity (PPP) per patient, while the use of the conventional service costs USD 99.94 PPP dollars per patient. As the performance of patient care was used as a parameter of effectiveness, there is no significant difference in effectiveness between the alternatives. If the services have equal patient care capacity, there are only differences in costs, which result in USD 52.59 PPP more for the care of a patient in the conventional strategy compared to telecardiology.	It is then concluded that conventional care is dominated by telecardiology, presenting itself as a good strategy to be implemented and/or financed by the public administration. This service will be able to contribute with greater agility in scheduling and carrying out consultations, preventing cardiovascular problems from evolving in severity, as well as helping people from regions farther away from the places where the specialists are located to have easier access to the related exams.

**Table 3 ijerph-18-10218-t003:** Results of Regula Mais Brazil Project.

Teleconsultation	Diseases	N (%)
Reason for teleconsultation	Insulin-dependent diabetes	80
	Non-insulin dependent diabetes	67
	Hypothyroidism/myxedema	4
	Hypertension with complications	3
	Thyreoid cancer	2
	Goiter	1
	Hypertension without complications	1
	Hyperthyroidism	1
	Other diseases	2
Technology used	Phone	46 (28.6%)
	Video calling	115 (71.4%)
Outcomes after the last teleconsultation	Follow up at primary care	40.3%
	Follow up at specialized care	33.8%
	Teleconsultation follow up	23.4%
	Urgent follow up at specialized care	2.6%
Net promoter score (NPS)	0–100	93

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
