# Peer review of "Description of e-Health Initiatives to Reduce Chronic Non-Communicable Disease Burden on Brazilian Health System"

_ijerph, 2021, doi:10.3390/ijerph181910218_

Round 1

Reviewer 1 Report

The paper is more a project report than a research report as written. The project is interesting but the theory that links the projects is lacking and the reader sees it as a statement of arguing for a change in health delivery rather than a scientific inqury. The projects are described separatedly, the methods to study the issues must be linked and the outcomes not only presented separately but also as a theoretical whole. The trial study is a feasibility  study rather than a full study and no power is accounted for in the method. In the conclusions the findings are presented as a argument for introducing new ways in public health and diagnostic medicine although they, by no means, alow such from a scientific base. I consider the content interesting but needs to be worked on to present a more true picture of the findings and their relevance for future research and development of new and more efficient health care approaches to Non-communicable Chronic diseases in Brazil. I agree that eHealth is a pomising addition to health care but it is not at the state of knowledge to overturne current face to face professional practice. To do so we need more evidence and critical approaches in testing these methods from a variety of viewpoints with a paptient as well as populationcentered focus.

The paper is written to convince the professional readers that telemedicine is as good as face to face services - this needs to be changed with a more humble and critical approach both in the introduction, methods, results and especially the discussion. There is plenty of literature of both and a more balanced presentation of the projects is warranted. I miss discussions on limitations and understanding of the interventions given and not given by the different methods and the title of the paper is pretty bold. It is a valiant aim but a risky thing to change the services by the vage evidence. No data is presented of fals negative or positive diagnosis etc. To make a suggestion that governments change the practice from these findings is not acceptable.

A interesting approach but needs to be edited from the perspective of safe practice and sound scientific frame of reference.

Author Response

Dear Reviewer,

Thanks for your notes. Indeed the main objective was to make a compilation of three strategies that are being done in the public health system in Brazil. Because they are different projects, we ad to describe the methods in different ways and the results also.

We didn´t insert more details regarding the Teleconsulta diabetes because the design paper was published before. In lines 228 and 229 we wrote an explanation about it.

Regarding the humble conclusion, we also agree. We write one paragraph about the limitations of the paper in lines 285-290.

Again, thank you very much for spending your precious time reading our paper.

Reviewer 2 Report

Dear Authors,

It was a pleasure reading your work. I totally agree that this is the least explored area of research. Your research therefore is important and timely, especially during this era of Covid 19.

While I found that you have narrated your story well, below I include some comments that may help improve your manuscript, if you decide to consider. I have outlined these under subheadings:

  • Title:
    • I have a question whether these are the only eHealth / telehealth initiatives offered in Brazil? If so, please mention this in your background / methodology. If not, I suggest that your title should read as follows: "The description of selective eHealth initiatives to reduce chronic non-communicable disease burden on Brazilian health system" Please note that you could shorten this title but it should give a clear picture that this is not a full evaluation of the initiatives (since project 3 is still underway) but a description of the selective few.
  • Abstract section:
    • For your statement on lines 24 and 25, you should be specific and indicate which project showed the "increased access to specialized care", which showed "reduced waiting time" and which showed "favoured distance education process".    
  • Introduction:
    • Please delete the statement on lines 48 to 51 as it is repeated on lines 51 to 54.
    • Please supply a reference for the statement on lines 55 and 56
  • Methods section:
    • Add the term "as" after "defined" on line 85 to make your statement reader friendly
    • To make it reader friendly, I suggest the title for Figure 1 should read as follows "Brazil states with telehealth initiatives included in the current research/article"  
    • To make it reader friendly, start the statement on line 103 with "Table 1 outlines the...
    • In Table 1, for "implementation analysis" under the "methodology section", Please indicate who was included for the semi-structured interviews? members OR implementers of the telehealth?
    • It would be helpful if you at least mention and describe the "monitoring of the waiting period process" covered under project 2 (i.e. Regula Mais Brasil Colaborativo)
    • The statements on lines 126 to 129 do not make sense. please revise them
    • Finally, overall, in the methods section it would be helpful if you describe or add a reference that take a reader to the description of all the different methodologies used to evaluate the projects (e.g. the logical model, socio-historical studies, cost-effective analyses etc.)
  • Results: 
    •  On line 151, what do you mean by saying "the project results offered subsidies"? Shouldn't you be saying "the project results highlighted opportunities for improvement"? OR am I not understanding your point?
    • The sentence on lines 151 to 156 is too long. Consider breaking it into shorter sentences that will be reader friendly.
    • The statement on line 156 does not read well. can you perhaps add "used, offered..." after the term "methods"?
    • Again, the sentence on lines 156 to 160 is too long. Consider breaking it into shorter sentences that will be reader friendly.
    • The term "analyzes" on line 157 should be "analyses"
    • on line 234 I am not sure what this (...) means
    • The sentence on lines 252 to 257 is too long
  • Discussion 
    • It would be helpful if a paragraph on limitations of this research is included. Especially that this is not a full project evaluation study and that project 3 is still ongoing.

Regards

Author Response

Dear Reviewer,

Thank you so much for all your comments!

We totally agree with changing the article´s title. Thank you for your suggestion.

The second suggestion on abstract and introduction, we made the correction. Thank you again.

All the suggestions and corrections were made accordingly to your orientations. Thank you again for your kindness and mentorship.
